

# Dependence of Predictability of Precipitation in the Northwestern Mediterranean Coastal Region on the Strength of Synoptic Control

Christian Keil[1], Lucie Chabert[1], Olivier Nuissier[2], and Laure Raynaud[2]

[1]Meteorologisches Institut, Ludwig-Maximilians-Universität, Munich, Germany
[2]CNRM (Météo-France & CNRS), 42 avenue G. Coriolis, 31057 Toulouse Cédex, France

**Correspondence:** Christian Keil (christian.keil@lmu.de)

**Abstract.** The weather regime dependent predictability of precipitation in the convection permitting kilometric scale AROME-EPS is examined for the entire HyMeX SOP1 employing the convective adjustment timescale. This diagnostic quantifies variations in synoptic forcing on precipitation and is associated with different precipitation characteristics, forecast skill and predictability. During strong synoptic control, which is dominating the weather on 80% of the days in the 2-months period, the domain integrated precipitation predictability assessed with the normalized ensemble standard deviation is above average, the wet bias is smaller and the forecast quality is generally better. In contrast, the spatial forecast quality of most intense precipitation in the afternoon, as quantified with its 95th percentiles, is superior during weakly forced synoptic regimes. The study also considers a prominent heavy precipitation event that occurred during the NAWDEX field campaign in the same region, and the predictability during this event is compared with the events that occurred during HyMeX. It is shown that the unconditional evaluation of precipitation widely parallels the strongly forced weather type evaluation and obscures forecast model characteristics typical for weak control.

## 1 Introduction

The Mediterranean region is affected by intense precipitation events every year particularly during the autumn months. Very high rain amounts and ensuing flash floods can cause widespread damage. Accurate prediction of these precipitation events is crucial to take precautions, warn the public and mitigate potential consequences. The HyMeX (Hydrological Cycle in the Mediterranean Experiment) field campaign was designed to advance the knowledge of Mediterranean heavy precipitation and flash-flooding events, to improve numerical models and to examine the representation and predictability of high-impact weather events (Ducrocq et al., 2014, and references therein).

Precipitation represents a very important yet challenging forecast variable due to the involvement of many atmospheric variables and the role of inherently highly non-linear processes in its formation. Forecasting precipitation with numerical weather prediction models requires among others a sufficiently fine model resolution to explicitly represent important processes like deep convection and a precise description of the microphysical processes leading to precipitation, but also an ensemble approach to quantify the forecast uncertainty.





In the last decade ensemble prediction at the convection permitting kilometric scale has become a standard technique for
weather forecasting and provides an important tool to forecast forecast uncertainty. However, the intermittency and spatiotemporal variability of precipitation on the kilometric scale renders the assessment of accuracy even more difficult. Next to the probabilistic approach an evaluation of high-resolution ensemble forecasts of precipitation calls for spatial measures to assess forecast quality.

The predictability of weather in general, and precipitation in particular, is weather regime dependent (Anthes, 1986; Bauer et al., 2015; Yano et al., 2018). The prediction of precipitation is influenced by the synoptic-scale environment and local processes and instabilities. Previous results suggest that there is higher forecast quality and above average predictability, that is lower uncertainty during strong synoptic control. The convective adjustment timescale offers an objective measure to classify weather regimes into strong and weak synoptic control on precipitation (e.g. Keil et al., 2014; Surcel et al., 2017). For instance, Schwartz and Sobash (2019) apply this diagnostic and conclude that forecast quality is related to forcing strength, with higher accuracy in more strongly forced regimes over the conterminous United States.

In the present study we aim to systematically identify different predictability regimes of precipitation in southeast France and northwest Italy during autumn 2012, for which the HyMeX campaign offers an unprecedented transnational observational dataset to validate convective scale ensemble prediction systems (Ducrocq et al., 2014). This period extends from 5 September to 5 November 2012 of which 59 days experienced noteworthy precipitation and includes numerous well studied IOPs of high impact weather situations. Here, forecasts of the kilometric scale AROME-EPS system are evaluated for the first time with neighbourhood methods to examine the spatial distribution of precipitation and to infer on different predictability levels conditional upon the weather regime of the day.

Previously Bouttier et al. (2016) evaluated the AROME-EPS system over the full HyMeX SOP1 period and identified strengths and weaknesses. The impact of initial conditions and model surface perturbations show a significant effect on the ensemble performance. Using a variety of conventional scores like RMS errors and spread-skill relation complemented with the probabilistic measures rank and ROC diagrams applied on near surface variables they found specifically for precipitation an almost negligible impact of direct surface perturbations and a lack of spread. The present study extends this comprehensive work, focuses on ensemble precipitation forecasts over the contiguous 2-months period and adds the aspect of weather regime dependent predictability.

Nuissier et al. (2016) presented a probabilistic evaluation of two convection permitting EPSs for the full HyMeX SOP1 and document a slightly better performance of AROME-EPS forecasts in terms of discriminating behaviour and reliability of 6-hourly rainfall. However, results depend on the choice of the verification domain since small areas suffer from sampling issues and the occurence of precipitation events. The examination of the HyMeX 'golden case' (IOP16a on 26 October 2012, Ducrocq et al., 2014) reveals slightly different predictability levels in two different subdomains in which precipitation is crucially governed by the location and deepening of a surface low pressure system over the Mediterranean Sea controlling the southerly moist low-level flow. Earlier, Hally et al. (2014) investigated the sensitivity of precipitation forecasts in an experimental convective-scale ensemble based on the Meso-NH model to diverse initial and boundary conditions and microphysical uncertainties for two IOPs (IOP6 and IOP7a in southeast France). Since both cases developed under strong synoptic forcing





the impact of atmospheric conditions on the spatiotemporal distribution of precipitation outweighs the one of surface conditions. It is suggested that the specific influence of surface conditions is larger for weakly forced events. Recently, Fourrié et al. (2019) revisited the HyMeX SOP1 period and demonstrate improved rainfall forecasts with a second reanalysis using 24% more additional data in the AROME system. The superior performance is specifically illustrated employing the conventional scores frequency bias and equitable threat score on 24h rainfall accumulations for an intense precipitation event that occurred over Spain and southern France on 29 September 2012 (IOP8).

Beyond HyMeX the downstream impact of synoptic systems on the predictability of high impact weather in the Mediterranean has been one of the science goals of the NAWDEX (The North Atlantic waveguide and downstream impact experiment) campaign in autumn 2016 (Schäfler et al., 2018). One of the NAWDEX highlights represents IOP9 on 13 October 2016 when the 24 h accumulated precipitation in southeast France reached 250 mm ahead of the cyclone Sanchez. This prominent case is included in the present study to be compared with the 2-months HyMeX SOP1.

This article focuses on precipitation predictability examining temporally highly resolved forecasts (3-hourly) of the AROME-EPS and relates differences in ensemble spread and forecast skill to broader environmental characteristics for the entire HyMeX SOP1 period. The remainder of the paper consists of a methods section, followed by a classification of the 2-months period into weather regimes, an illustration of three prominent cases, the verification using classical gridpoint based quality measures, probabilistic metrics and a spatial score to allow for location tolerance, and finishes with conclusions.

## 2 Model, Data and Metrics

### 2.1 The convective-scale ensemble

The AROME-EPS used in this present study is based on the AROME forecasting system largely described in Seity et al. (2011) and in Brousseau et al. (2016). It is based on adiabatic, non-hydrostatic equations from the limited-area ALADIN (Aire Limitée Adaptation dynamique Développement InterNational) model (Bénard, 2004; Bubnova et al., 1995). A horizontal resolution of 2.5 km and 60 (HyMeX ensemble) or 90 (Nawdex ensemble) vertical levels are used in this study. AROME shares the same physical parameterizations as the research model Meso-NH (Lac et al., 2018), including a bulk one-moment microphysics scheme following Caniaux et al. (1994), which represents six water species (water vapour, cloud water, rain water, primary ice, graupel and snow). The representation of the turbulence in the planetary boundary layer is based on a prognostic turbulent kinetic energy (TKE) equation combined with a diagnostic mixing length (Bougeault and Lacarrère, 1989). The TKE scheme used in AROME was developed by Cuxart et al. (2000) and the scheme is derived from the full set of equations for second-order moments. At 2.5-km resolution, the deep convection is assumed to be explicitly resolved by the model's dynamics. However the shallow convection requires a parameterization of subgrid effect for which the Pergaud et al. (2009) scheme is used. It is a mass flux scheme based on the eddy diffusivity mass flux (EDMF) scheme (Soares et al., 2004) that parameterizes dry thermals and shallow cumuli.

The AROME-EPS ensemble setup is the following: a) The ensemble comprises 12 members. For the HyMeX period ensemble simulations start at 00:00 UTC (up to 36-h forecast range), whereas ensemble runs are initialized at 21:00 UTC (up



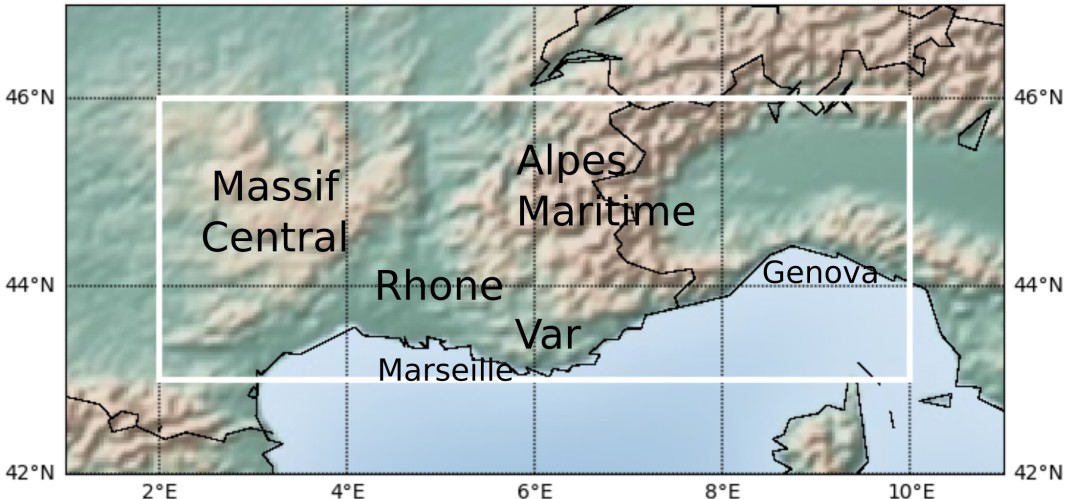

**Figure 1.** Map illustrating the Northwestern Mediterranean domain (white box) and geographical landmarks used in the text.

to 45-h forecast range) for the Nawdex case. b) In the ensemble simulations, AROME is driven by the global short-range ARPEGE-EPS (Descamps et al., 2014), called hereafter PEARP. Firstly, a subset of 12 members of the PEARP is selected according to the Nuissier et al. (2012) technique. The PEARP 35-member ensemble forecasts are classified by a complete-linkage

clustering technique (Molteni et al., 2001). c) The initial conditions are provided by adding downscaled forecast perturbations of the selected PEARP members to the AROME operational analysis (Raynaud and Bouttier, 2017). d) Atmospheric model errors are represented through the so-called SPPT scheme (stochastic perturbation of physics tendencies) described in Bouttier et al. (2012), which simulates the effect of random errors due to the physical parametrizations. e) Finally, random perturbations are added to various parameters of the surface externalisée (SURFEX) surface scheme, including for instance sea-surface

temperature, soil moisture and temperature perturbations (Bouttier et al., 2016).

## 2.2   Domain and observational data

The investigation domain extends across $300\,\mathrm{km} \times 800\,\mathrm{km}$ and encompasses southeastern France and northwestern Italy including the coastal regions of Cote d'Azur and Riviera as well as adjacent mountainous regions of the Massif Central and the Alpes Maritimes (Fig. 1). This region, that is herein called the Northwestern Mediterranean, is prone to heavy precipitation

generated by a wide variety of flow conditions including synoptic systems characteristic of Rossby wave breaking at the eastern end of the North Atlantic storm track, modulated by orography and thermal contrasts of the Mediterranean basin as well as calm, conditionally unstable situations requiring trigger mechanisms to generate rainfall (e.g. Ducrocq et al., 2014; Nuissier et al., 2016, and references therein). The choice of the location and size of the investigation domain is carefully chosen and represents a compromise between being large enough to have numerous precipitation events giving good statistics, but small

enough to comprise a specific and unambiguous meteorological situation in combination with the good coverage of rainfall





observations in the Northwestern Mediterranean. If the domain is too large strongly differing meteorological systems may be contained and the results obtained using area averages may be blurred and not representative. However, we believe that the chosen domain encompassing $300\,\mathrm{km} \times 800\,\mathrm{km}$ represents a good compromise being at the scale of the Rossby radius of deformation. The domain size conforms with the recommendation of Wernli et al. (2009) to use areas smaller than $500\,\mathrm{km} \times$

$500\,\mathrm{km}$ to compute an unequivocal spatial forecast quality value representative of a certain meteorological situation.

As observational data we use 3-hourly rain-gauge observations retrieved from the HyMeX database and hourly rain-gauge observations for the NAWDEX case that are accumulated 3-hourly. The rain-gauge observations are spatially interpolated to the model grid using a linear barycentric interpolation to perform the spatial evaluation.

### 2.3 Metrics and measures

Generally, an ensemble of forecasts provides a range of possible scenarios allowing for the estimation of forecast uncertainty. Large deviations of individual ensemble members point towards large forecast uncertainty and low predictability. One method to estimate the predictability of precipitation is the computation of the normalized standard deviation $S_n$ (e.g. Hohenegger et al., 2006; Nuissier et al., 2016). In the present study $S_n$ is calculated at any gridpoint where the 3-hourly precipitation rate exceeds $1mm(3h)^{-1}$, and is subsequently area averaged. Larger $S_n$ values indicate higher ensemble dispersion, larger forecast

uncertainty and lower predictability.

A hierarchy of measures is applied to conduct the weather regime dependent verification of precipitation forecasts during the HyMeX SOP1. Following the gridpoint based spread (STDEV) and root mean square error (RMSE) we present two probabilistic measures Relative Operating Characteristics ROC and reliability diagram complemented with the widely used Fractions Skill Score (FSS, Roberts and Lean, 2008) to account for spatial tolerance.

### 2.4 The convective adjustment timescale

The convective adjustment timescale $\tau_c$ constitutes an objective measure to classify weather situations by taking the ratio of convective instability (measured by CAPE) and its removal (expressed by the precipitation rate; Done et al., 2006; Keil et al., 2014). During synoptically forced weather precipitation balances the production of instability generated by, e.g., large-scale ascent, the atmosphere is in equilibrium and the value of the convective adjustment timescale is small (Zimmer et al., 2011). In

contrast, when the synoptic forcing is weak, local processes like solar insolation, or the interaction with orography generating convergence lines are necessary to overcome a barrier and release convection. During this non-equilibrium regime large CAPE values can build up before convection is triggered and the convective adjustment timescale amounts to larger values comparable to the synoptic timescale.

The area averaged $\tau_c$ value can be used to categorically determine the weather regime of the day (e.g. Kühnlein et al., 2014;

Keil et al., 2019). Firstly, Gaussian smoothed forecast fields of 3-hourly precipitation rates and most unstable CAPE are taken to calculate $\tau_c$ at any gridpoint exceeding $3mm(3h)^{-1}$ in individual members given that a minimum areal coverage of this precipitation rate is reached (consistent with the thresholds applied in Kühnlein et al., 2014). Secondly, the ensemble mean of individual $\tau_c$ values at any gridpoint is computed, and thirdly, the domain average of this ensemble mean is taken. Finally, if



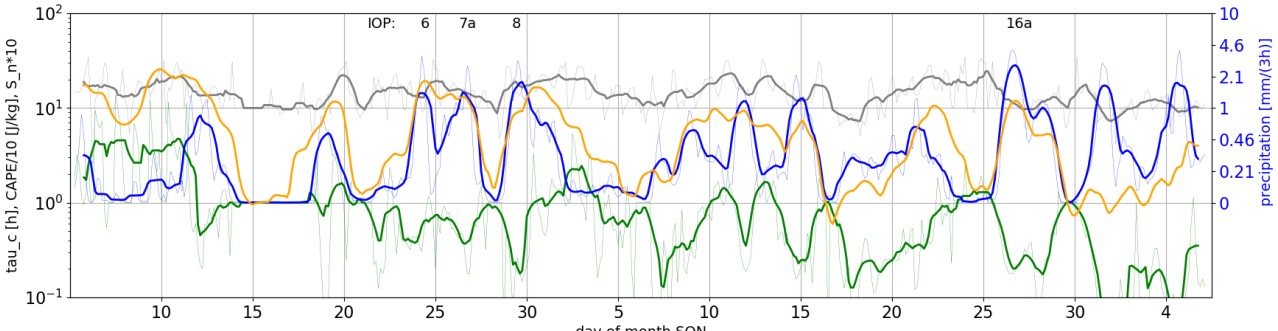

**Figure 2.** Time series of forecasted area averaged ensemble mean precipitation (blue), convective adjustment timescale ($\tau_c$, green), CAPE (orange), and normalized spread of precipitation ($S_n$, gray) in the Northwestern Mediterranean Coastal domain for the entire SOP1 period. The data is plotted using a 24-hourly moving average on the 3-hourly values to increase readability. The thin lines represent the 3-hourly data of precipitation (blue), the convective adjustment timescale (green) and the normalized spread of precipitation ($S_n$, gray) for reference. Note that the dependent variables are given in logarithmic scale, CAPE is divided by a factor of 10, $S_n$ multiplied by a factor of 10 and precipitation is labeled on the right hand side.

the maximum domain averaged ensemble mean $\tau_c$ exceeds a threshold criterion at least once a day, that day is classified to be

145 weakly forced.

In the present study a threshold value of 3 h is chosen to account for the smaller $\tau_c$ values occurring in the autumn season of HyMeX SOP1 (see Fig. 3). This is in contrast to applications of $\tau_c$ in the summer season when a threshold of 6 h is reasonable to separate mid-latitude precipitation regimes due to dynamic control (Kühnlein et al., 2014; Keil et al., 2019). However, Zimmer et al. (2011) argue that the $\tau_c$ diagnostic results in a continuous distribution and conclude that a value somewhere between 3

150 and 12 h clearly distinguishes between different regimes. It turns out that a threshold of 3 h substantially reduces the sampling error giving a distribution of 48 strongly vs 11 weakly forced days during HyMeX SOP1.

## 3 Classification based on the strength of synoptic control

At first glance the timeseries spanning the entire 2-months period shows the variability of weather on daily timescales in autumn 2012 (Fig. 2). The precipitation curve highlights some of the 'golden cases' observed during HyMeX SOP1 (e.g. IOP6

155 on 24 Sep, IOP7a on 26 Sep, IOP8 on 29 Sep, IOP16a on 26 Oct) with peaks exceeding $2mm(3h)^{-1}$ in domain integrated precipitation. The timeseries of convective instability (CAPE) exhibits large variations, too. High values of spatially averaged CAPE exceeding 100 J/kg mostly concur with the occurence of strong precipitation events (e.g. IOPs 6, 7a, 8 and 16a) pointing towards their predominantly convective character, whereas sometimes maxima do not coincide (e.g. beginning of September). During these episodes convective instability is created by, for instance, solar insolation but cannot be removed by precipitation

160 because of inhibiting factors like capping inversions atop the boundary layer prevent convection initiation. The rank correlation





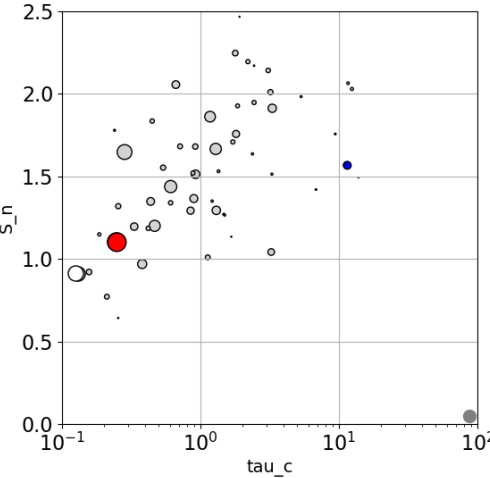

**Figure 3.** Scatterplot of domain averaged, daily maximum convective adjustment timescale and daily mean normalized standard deviation of precipitation for the entire SOP1. The prominent cases discussed in Section 4 are highlighted: the red circle represents the HyMeX IOP16a, the blue circle the HyMeX 11 September 2012, and the white circle the NAWDEX case. The size of the symbols indicates the daily precipitation accumulation, with the gray circle in the bottom right corner displaying a daily domain integrated rainfall accumulation of 10 mm for comparison.

of CAPE and 3-hourly precipitation (and its normalized standard deviation $S_n$) amounts to $0.44$ (and $0.28$, respectively) and confirms the limited predictive power of CAPE alone.

Here the convective adjustment timescale $\tau_c$ provides a better suitable measure to distinguish and to classify weather situations with different synoptic control. Using a categorical threshold of $3\,\mathrm{h}$ for the daily maximum area averaged convective adjustment timescale results in 48 strongly and 11 weakly forced days in the Northwestern Mediterranean domain during HyMeX SOP1. Many of the weakly forced cases occur in the first week of the SOP1 (8 to 11 Sept, Fig. 2). After mid-October there are no weakly forced cases anymore suggesting the influence of the seasonal cycle, as decreased solar insolation limits diurnally-driven precipitation. However it is the interplay between the creation of convective instability and its removal by precipitation (both variables make $\tau_c$) that shows the overall decrease in autumn that is strongly modulated by the occurrence of mid-latitude weather systems. During SOP1 $\tau_c$ exceeds the threshold value ultimately on 13 October, while area averaged CAPE maxima exceeding 100 J/kg still occur in late October (e.g. on 26 October, IOP16a). A comparison of the timeseries of $\tau_c$ and of the normalized standard deviation $S_n$ in Fig. 2 gives a first indication of a connection between both, that is between the weather regime and the forecast uncertainty. Large values of $\tau_c$ indicating weakly forced weather conditions correspond with above average values of $S_n$ suggesting below average precipitation predictability.

This relationship and clear dependence of the convective adjustment timescale $\tau_c$ and the normalized standard deviation $S_n$ of precipitation is further illustrated in Fig. 3. Large values of $\tau_c$ correspond with large $S_n$ of precipitation being a sign of





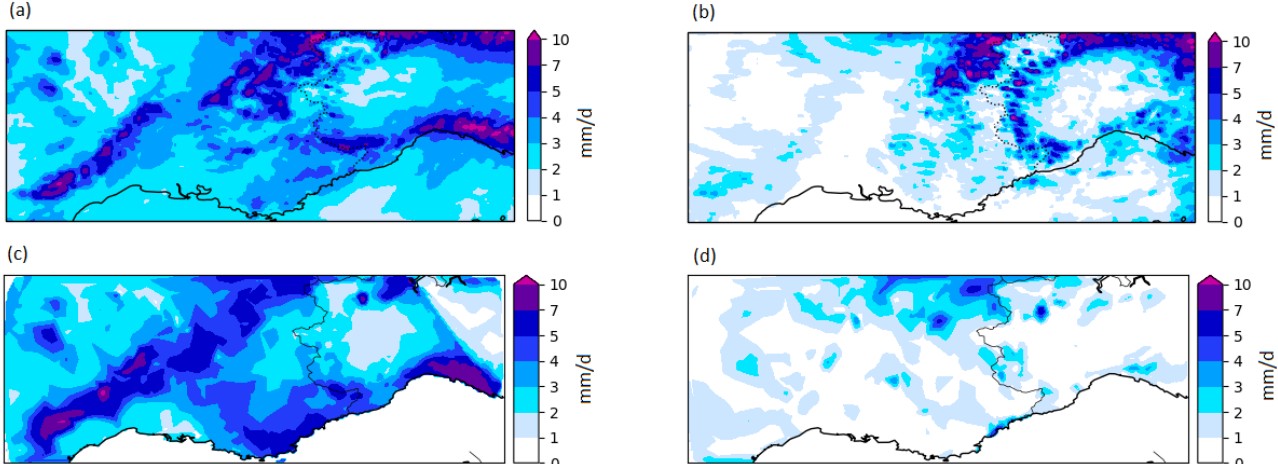

**Figure 4.** Aggregated mean daily precipitation divided into strong (a,c) and weak (b,d) forcing conditions. Displayed are ensemble mean (a,b) and interpolated rain-gauge observations (c,d) of 24 h precipitation for the 2-months period in autumn 2012.

below average predictability. The strongest precipitation events (e.g. IOP16a and the NAWDEX case) occur predominantly at low $\tau_c$ values when the normalized ensemble spread of precipitation ($S_n$) is small, too. Thus, in a domain integrated sense, synoptically forced situations cause higher precipitation accumulations with lower forecast uncertainty. The rank correlation

between $\tau_c$ and $S_n$ is 0.6 providing statistical evidence that $\tau_c$ can be reasonably used as a predictor to classify weather regimes with inherently different precipitation predictability. Moreover, the scatterplot shows that the majority of $\tau_c$ values amounts to less than 3 h and a comparison with Fig. 4 in Keil et al. (2019) confirms that the chosen threshold value represents a sensible classification criterion in this specific region at that time of the year.

The different dynamical control shows its fingerprint in the mean spatial distribution of daily rainfall, too (Fig. 4). Apparently,

there is more precipitation during strongly forced weather situations than during weakly forced ones. The regions receiving more than 5mm/24h during synoptic control at the southeastern foothills if the Massif Central, the western foothills of the Alpes Maritimes and the Mediterranean coast East of Marseille agree well with observations (Fig. 4a,c). In contrast, a spottier distribution of daily rainfall concurrent with an overestimation of precipitation totals becomes evident during weak synoptic control (Fig. 4b,d, and later in Fig. 8).

The aggregated diurnal evolution during weak synoptic control shows the characteristic behaviour with a pronounced diurnal cycle of $\tau_c$ peaking in the early afternoon shortly before maximum precipitation rates occur in the late afternoon (Fig. 5). Conversely, during strong forcing conditions there is almost no diurnal pattern in $\tau_c$, precipitation rates are higher and show a weaker amplitude with maxima in the early evening. The unconditional average of $\tau_c$ and precipitation is fairly similar to the strongly forced weather type because these flow conditions are prevalent during the HyMeX period thus dominating the

diurnal evolution.



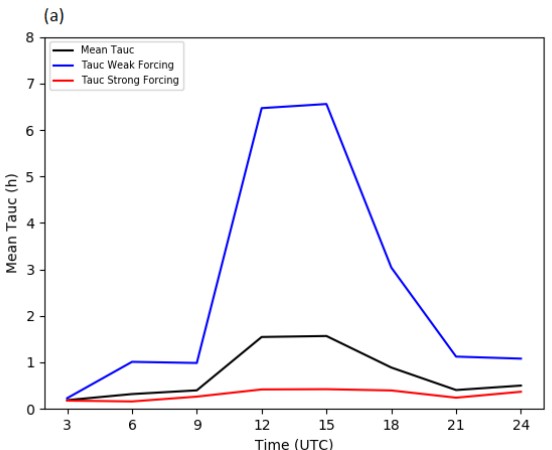
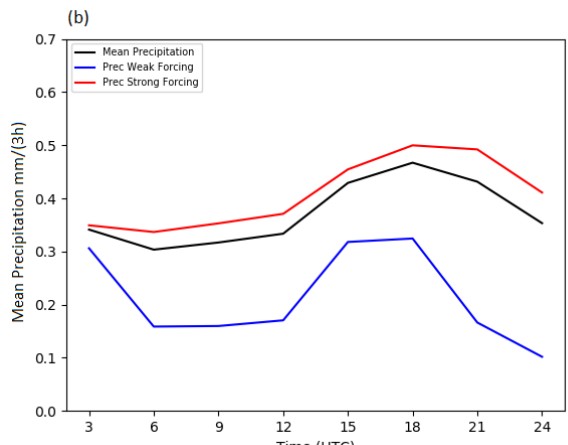

**Figure 5.** Aggregated diurnal cycle of the ensemble mean convective adjustment time-scale (a) and 3-hourly precipitation (b) averaged over the full SOP1 period (black) and over weakly (blue) and strongly forced weather regimes (red) across the Northwestern Mediterranean domain.

## 4 Three prominent and representative cases

In this section three characteristic cases are presented to highlight the different nature of individual events in detail and to identify hypotheses to be proven in the subsequent systematic evaluation using a hierarchy of measures. The prominent cases comprise HyMeX IOP16a (26 Oct 2012), a typical weakly forced situation during HyMeX (11 Sept 2012) and the NAWDEX Sanchez case (13 Oct 2016). The daily timeseries of precipitation, CAPE and $\tau_c$ clearly depict their different character (Fig. 6). On IOP16a and the NAWDEX case $\tau_c$ is always considerably smaller than the threshold criterion (and even less than 1 h, Fig. 6a,c) indicating strong synoptic control (as for IOPs 6, 7a; not shown) in agreement with Hally et al. (2014) and Nuissier et al. (2016), whereas the temporal evolution of precipitation, CAPE and $\tau_c$ on 11 Sept 2012 shows the characteristic behaviour of weakly forced weather situations (Fig. 6b), that is high $\tau_c$ values preceeding the strongest rainfall (compare to Keil et al. (2014)).

### 4.1 Strongly forced case on 26 October 2012 (IOP16a)

HyMeX IOP16a represents a case of deep convection that developed over the western Mediterranean Sea and affected the coastal regions of France and Italy. The synoptic situation was characterized by a deep upper-level low centered over the Iberian Peninsula moving slowly eastward and fueling slowly propagating Mesoscale Convective Systems in the Northwestern Mediterranean. IOP16a represents a "golden case" enabling us to address the predictability of a high impact weather event (Ducrocq et al., 2014; Nuissier et al., 2016).





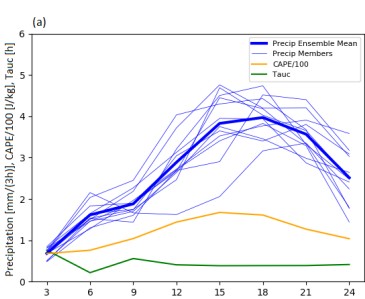 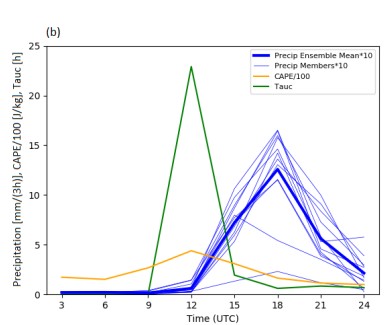 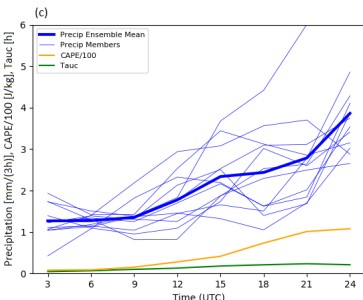

**Figure 6.** Time series of ensemble mean area averaged precipitation, CAPE and convective adjustment timescale $\tau_c$ for the prominent cases (a) HyMeX IOP16a, (b) HyMeX 11 Sep 2012 and (c) NAWDEX 13 Oct 2016. Additionally the precipitation timeseries of the individual ensemble members highlights intra-ensemble variability. Note the different scale on 11 Sept 2012.

Fig. 7 depicts the spatial distributions of 6 h ensemble mean precipitation, its intra-ensemble variability and four individual ensemble members (all 12 to 18 UTC). Three hotspots of precipitation are forecast on 26 October. The spatially largest is located across the Massif Central, where the ensemble mean exceeds 10mm/6h, the intra-ensemble variability (represented

by $S_n$) is small and all members indicate widespread precipitation. At the southeastern foothills of the Massif Central in the Cevennes region the ensemble overestimates the 6 h precipitation (exceeding 20mm/6h) and there is a considerable ensemble spread. There, forecasts of single members diverge (Fig. 7c-f) and show a displacement of heaviest precipitation (e.g. shifted eastward in member 7, very intense and southward in member 8). In the eastern Rhone valley, an area where the ensemble mean indicates more than 5 mm, the intra-ensemble variability is large and individual members fail to predict any precipitation

(e.g. member 12).

A second hotspot of strong precipitation occurs in the Var region where maximum rainfall accumulations are observed (larger 50mm/6h). There, larger values of $S_n$ point towards a higher intra-ensemble variability that becomes apparent when looking at the rainfall sums of individual members (e.g. Fig. 7e,f). A third heavy precipitation region is forecast close to Genova in Italy. Observations indicate a considerable overprediction in this region with filled circles depicting rain-gauge observations being

clearly recognizable (Fig. 7a). However, the hidden circles across large regions of the Massif Central and the Alpes Maritimes point towards the overall good performance of the ensemble mean forecast of precipitation. Interestingly precipitation amounts under strong synoptic control are strongly modulated by orography with highest intra-ensemble variability in the flat regions of the Rhone valley and south of the Massif Central.

## 4.2 Weakly forced case on 11 September 2012

Weather on 11 September represents a characteristic case of a weakly forced situation during HyMeX. Before noon single convective cells are triggered with very different intensity and location in the individual members (not shown) resulting in very small area averaged rainfall accumulations (Fig. 6). Subsequently, convection intensifies leading to more than 50mm/6h



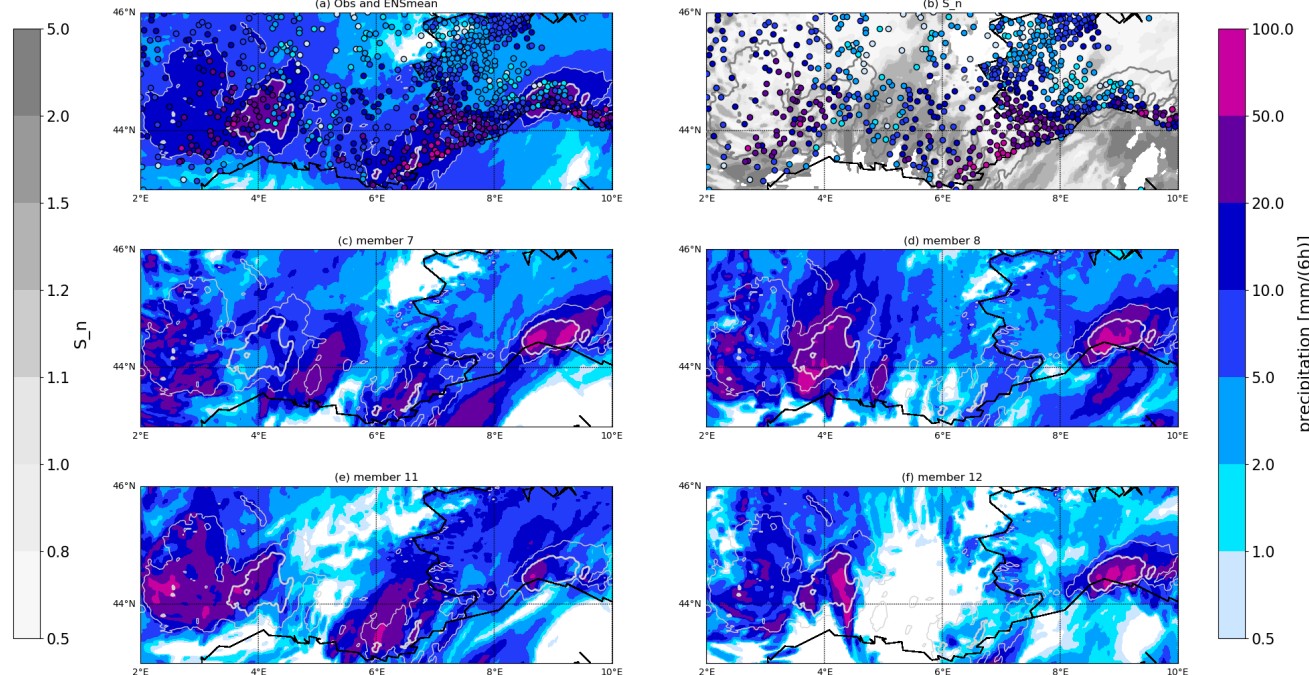

**Figure 7.** Illustration of 6-hourly precipitation for IOP16a valid 26 October 18 UTC: (a) ensemble mean precipitation forecast (initialized 26 October 00 UTC) and observations (filled circles), (b) normalized ensemble spread (gray) and rainfall observations (filled circles) for reference. Panels (c, d, e and f) show 6-hourly precipitation of selected individual ensemble members, all overlaid with 10 (thin) and 20 mm (thick line) ensemble mean precipitation contours.

in individual members at different locations (Fig. 8c,d,f). The differences in terms of exact location of heaviest precipitation result in maximum ensemble mean values of less than 20 mm (Fig. 8a). Overall, precipitation is strongest across mountainous
regions (a more or less distinct precipitation band extends from southwest to northeast across the Massif Central in all members) with correctly forecast dry conditions south of 44°N in the Var region. Whereas the accumulated precipitation distribution in members 1 and 3 resembles the ensemble mean pattern, other members exhibit big deviations: member 4 forecasts hardly any rainfall, while member 9 forecasts a lot of precipitation in the western part of the domain only. Large $S_n$ values west of 6°E demonstrate this considerable intra-ensemble variability. Overall, the comparison with rainfall observations suggests a notable
overestimation (Fig. 8a) and a clear connection to orography during weak control.

The weather situation on 11 September is characteristic for the first week of the HyMeX SOP1 period (see Fig. 2) when solar insolation in early autumn is still strong enough to generate convective instability by surface heating resulting in large



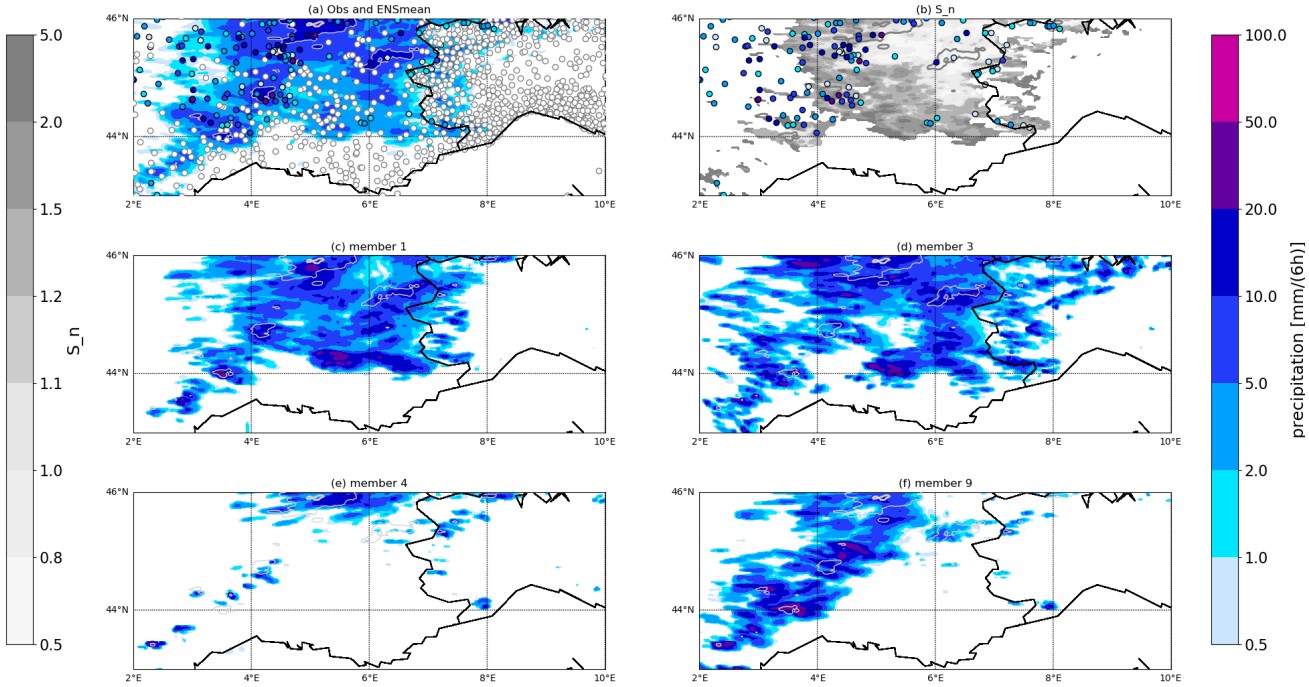

**Figure 8.** Same as Fig. 7, but for 11 September 18 UTC (initialized 11 September 00 UTC).

CAPE and large $\tau_c$ values indicating a need for local triggering mechanisms to overcome convective inhibition and to form precipitation (see Fig. 6).

## 4.3 Heavy precipitation on 13 October 2016 during NAWDEX

The detailed examination of individual heavy precipitation events in the Western Mediterranean region is complemented with one of the most prominent cases in that region that developed downstream of the cyclone Sanchez during the NAWDEX field campaign in autumn 2016 (Schäfler et al., 2018). Fig. 9 shows a good match of forecast and observed 24 h precipitation peaking in the Cevennes region with more than 200 mm. This event is clearly classified as strongly forced regime with an area averaged maximum $\tau_c$ of less than 30 min (see Fig. 3 and Fig. 6).

However, and unexpectedly for a case under strong control, individual ensemble members show surprisingly large spatial variability that becomes evident when inspecting the time window of heavy rainfall in the afternoon (between 12 and 18 UTC, Fig. 10). Focussing on the region of heaviest precipitation (20mm/6h in ensemble mean) at the southern foothills of the Massif Central all members exhibit strong precipitation rates individually, whereas in other areas there are large discrepancies (e.g.




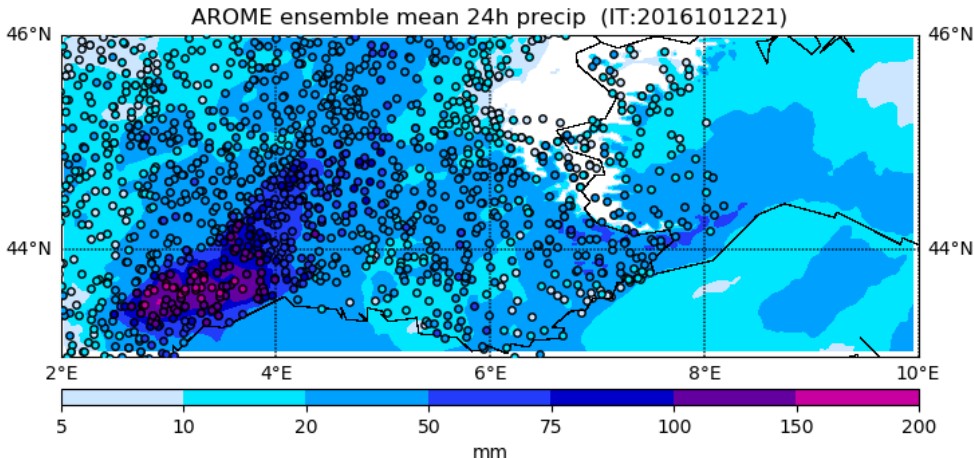

**Figure 9.** Observed (circles) and forecast (initialized 12 October 21 UTC) ensemble mean accumulated daily precipitation until 14 October 2016 6 UTC.

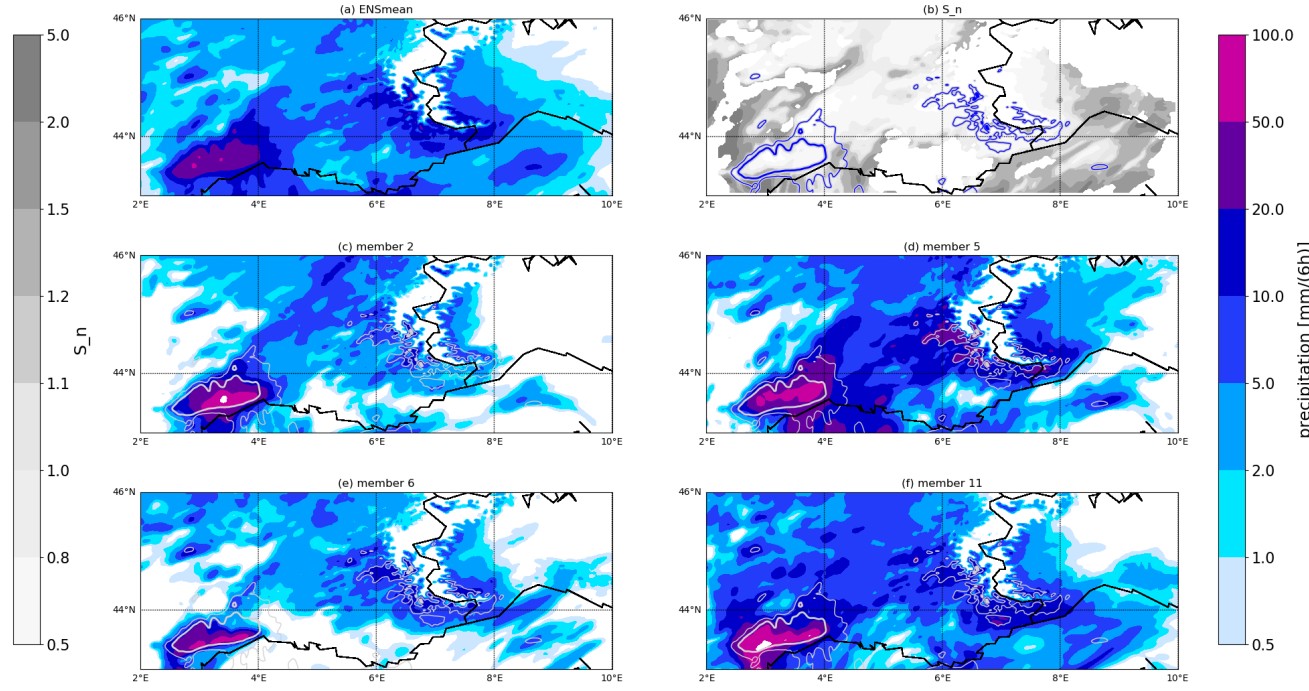

**Figure 10.** Same as Fig. 7, forecast for 13 October 2016 18 UTC (initialized 12 October 21 UTC).





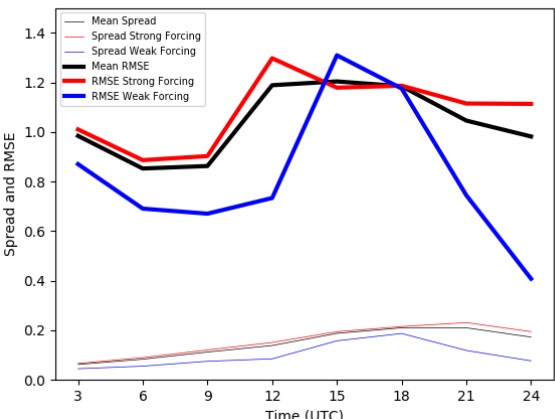

**Figure 11.** Aggregated time series of the spread (standard deviation) and skill (RMSE) averaged over the full SOP1 period (black) and over weakly (blue) and strongly forced weather regimes (red). The RMSE is the ensemble mean of the member RMS forecast error.

members 5 and 6, Fig. 10d,e). Within the heaviest precipitation region all members agree well resulting in small $S_n$ values. Above average precipitation variability occurs northward (across the Massif Central) and across the Mediterranean Sea.

In summary, the three selected cases indicate that the heaviest precipitation is co-located with orography during both regimes, that the spatial predictability of precipitation can considerably vary from case-to-case even within one forcing type, and that the precipitation intensity is overestimated during weak control.

**5   Systematic verification conditional to the strength of synoptic control**

Finally, we examine the question how precipitation forecasts during the different weather regimes compare with observations using a hierarchy of measures applied on the full 2-months period. Firstly, we show the mean diurnal evolution of the gridpoint based root-mean-square error (RMSE) of 3-hourly precipitation forecasts and rain-gauge observations conditionally averaged on both weather regimes.

During strong control the RMSE exhibits less diurnal variations than during weak control when a typical diurnal cycle is recognizable attaining the highest error during the convective most active period in the afternoon between 12 and 18 UTC (Fig. 11). The magnitude of the error reaches values up to $1.2 mm (3h)^{-1}$ in the Northwestern Mediterranean, which is roughly 50% less than found by Bouttier et al. (2016) looking at large parts of Western Europe. Given that rainfall rates during weak forcing amount to only about 60% of the rates during strong forcing (Fig. 5b), the relative error is higher in the weak regime.

Likewise, the ensemble spread shows a diurnal cycle and is highest during the convective most active period in the afternoon under weak synoptic control. Since 80% of the days during HyMeX SOP1 are classified as strongly forced weather regime it

is not surprising that the regime independent curve follows the strongly forced curve closely thus obscuring the forecast model characteristic during weak control.

Secondly, the regime dependent probabilistic performance of the ensemble is investigated using the ROC and reliability diagrams for 3-hourly (Fig. 12a,b) and daily accumulations (Fig. 12c,d). Both probabilistic scores highlight the superior performance during strongly forced weather regimes. A ROC curve closer to the left and upper boundaries displays a greater event discrimination in this weather situation. The larger distance of the ROC curve points during strong control indicates the higher *absolute* spread when 3-hourly (and daily) precipitation accumulations are averaged over the entire SOP1. In this weather regime the AROME-EPS forecasts are generally more reliable, in particular when averaged over 24 hours (Fig. 12d). The calibration functions in the reliability diagrams show that the forecast probabilities are consistently too large relative to the conditional observed relative frequencies. This is an indication of overforecasting equivalent to a wet bias. The general wet bias is strongest during weak synoptic control for short (3-hourly) time windows (Fig. 12b). Moreover, the flatness of the calibration function for this weather regime reveals a poor resolution and an overconfidence. Observed relative frequencies depend only slightly on the forecast probabilities and always amount to less than 20% for all forecast probabilities of moderate $3mm(3h)^{-1}$ precipitation rates. Relaxing the temporal exactness and extending the window to daily accumulations improves the reliability, in particular during strong control (Fig. 12d).

Finally, the Fractions Skill Score (Roberts and Lean, 2008; Faggian et al., 2014) is employed to address the double penalty problem inherent in convective scale precipitation forecasts. In Fig. 13 the ensemble mean FSS is shown as a function of neighborhood size for absolute rainfall rates ($0.3mm(3h)^{-1}$, a threshold frequently used to separate rain versus no-rain areas, and $10mm(24h)^{-1}$ accumulation) splitted into weather regimes aggregated for the 2-months period. During strong forcing the spatial forecast quality of the low rainfall threshold is superior for all neighborhood sizes (Fig. 13a). The skill increases when relaxing the grid point proximity and comparing larger neighborhoods, as expected. The fairly large box sizes and the extension of the whiskers demonstrate the large variability of forecast quality during the 2-months period. The upper quartile of the boxplot is touching upon a FSS value of 0.5 at neighborhood sizes of 125 km during strong forcing. A FSS value of roughly 0.5 is also known as the believable scale ($FSS = 0.5 + f_0/2$, where $f_0$ is the observed precipitation coverage, see Dey et al., 2014), a scale at which forecasts are deemed reasonably skillful and useful (Roberts and Lean, 2008). Thus, 25% of the time (3-hourly intervals on 48 strongly forced days, i.e. for 96 time windows within SOP1) the forecasts are skillful at a scale of $\mathcal{O}(100\,\mathrm{km})$, which is of the same order as found in previous studies (Clark et al., 2010; Mittermaier et al., 2011; Schwartz and Sobash, 2019; Bachmann et al., 2020), based on FSS and other neighborhood methods. Useful precipitation forecasts are hardly encountered during weak forcing using absolute rainfall rates.

Relaxing the temporal exactness of 3-hourly accumulations towards daily sums confirms previous results. Using a fixed precipitation amount of 10 mm per day reveals that the mean FSS during strong control always lies higher than during weak control, that is on average the spatial forecast accuracy is higher during strongly forced weather situations, as expected (Fig. 13b). The discrepancy between the mean and the median of FSSs during strong forcing suggests that the high threshold of $10mm(24h)^{-1}$ represents rare events with different intermittency characteristics in forecast and observation leading to a skewed distribution.

**Figure 12.** Relative Operating Characteristics (ROC) (a,c) and reliability diagram (b,d) for $3mm(3h)^{-1}$ precipitation rates (a,b) and daily amounts of $10mm(24h)^{-1}$ (c,d) conditional to weather regime aggregated over the entire HyMeX SOP1.





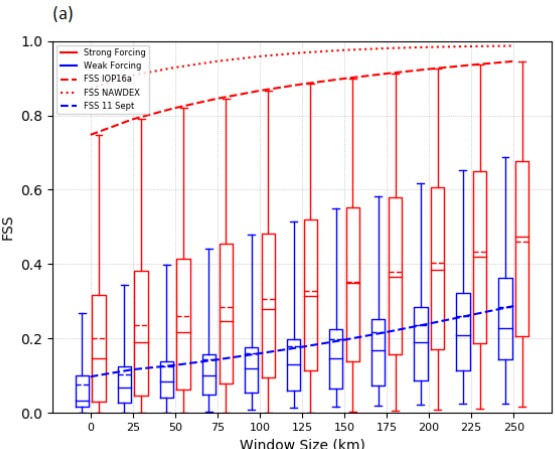
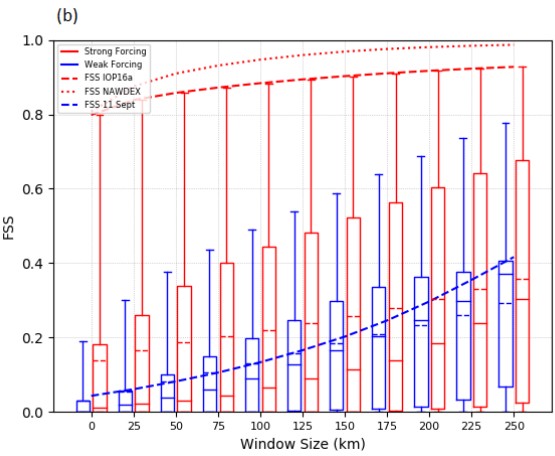

**Figure 13.** Fraction Skill Score FSS of ensemble mean 3-hourly precipitation vs interpolated rain-gauge observations as a function of neighborhood size conditional to weather regime: (a) $0.3mm(3h)^{-1}$ and (b) $10mm(24h)^{-1}$, all averaged for the entire HyMeX SOP1. The horizontal line (dashed line) within the boxes indicates the median (mean), respectively. The boxes and whiskers are slightly displaced at the discrete window sizes (weak to the left, strong to the right) to increase readability. Additionally, the FSS of the prominent cases is depicted.

The inspection of the spatial forecast accuracy of the prominent cases again highlights the large day-to-day variability. Both strongly forced prominent events (IOP16a and NAWDEX) exhibit a very good spatial forecast quality with the FSS reaching values larger 0.8 for window sizes larger 25 km tantamount with the highest whiskers (Fig. 13). The NAWDEX case (occurring
in 2016) even shows FSS values higher than the highest whiskers found during HyMeX. The excellent forecast performance is mainly caused by the low precipitation threshold ($0.3mm(3h)^{-1}$) and the widespread precipitation occurring on both days. Large parts of the domain receive such precipitation rates and the FSS attains high values. The prominent weakly forced case indicates an average forecast performance (FSS of 11 Sept matches the mean value of this regime) for low rainfall rates separating essentially rain and no-rain regions.
However, taking into account a varying model bias during different weather regimes changes the picture. The pure forecast location accuracy neglecting a model bias can be estimated by using percentiles of forecast and observed precipitation amounts. Whereas the 95th percentiles of forecast and observed precipitation agree well during strong forcing (at least until 18 UTC), there is a considerable overprediction during weak forcing (Fig. 14a). This overforecasting is strongest during the convective most active period in the afternoon between 12 and 18 UTC. Taking this bias into account by using precipitation percentiles
results in a superior spatial forecast quality during weakly forced regimes (Fig. 14b). Thus forecasting the location of heaviest precipitation in the afternoon (expressed by the 95th percentiles) is better during comparably quiescent synoptic-scale atmospheric conditions. This is at first sight an unexpected and surprising result. Given the favourable meteorological ingredients for generating heavy precipitation at this specific geographical region in the autumn season (Grazzini et al., 2020), we hypothesize that well represented steady land surface structures (like orography, particularly) in kilometric scale models provide sufficient





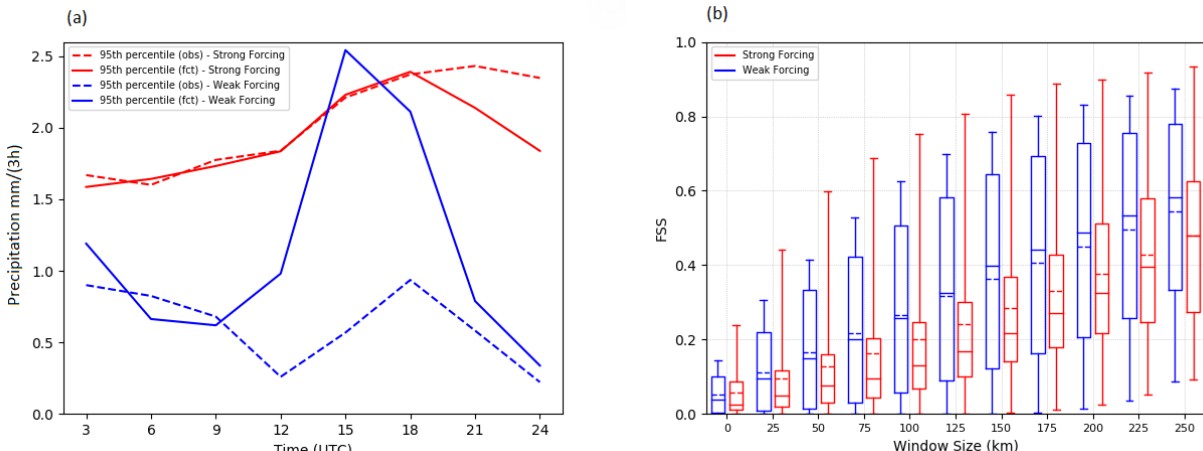

**Figure 14.** (a) Aggregated diurnal cycle of 3-hourly precipitation values corresponding to the 95th percentiles, both stratified in weakly (blue) and strongly forced weather regimes (red) and averaged over the full SOP1 period. (b) FSS of 95th percentiles of 3-hourly precipitation as a function of neighborhood size conditional to weather regime averaged for the entire HyMeX SOP1 between 12 and 18 UTC.

trigger mechanisms to initiate convection and serve as a permanent source of precipitation predictability during weak control. The structuring effect of mountains on the location of precipitation has previously been shown in idealized and real-world ensemble simulations of summertime convection in Central Europe (Bachmann et al., 2019, 2020). In contrast, forecasting the location of heaviest precipitation with high temporal exactness at forecast horizons of 12 to 24 hours is challenging during transient synoptic-scale weather systems typical during strong control.

**6    Conclusions**

This study extends prior work documenting the performance of AROME-EPS during HyMeX SOP1 (Bouttier et al., 2016; Nuissier et al., 2016) by the weather regime dependent aspect of precipitation predictability with a special focus on the spatial forecast quality. The convective adjustment timescale $\tau_c$ is used to categorically classify every single day within the 2-months period in autumn 2012 into one specific weather type depending on the strength of the synoptic control. From a physical

perspective, it is sensible to use variations in forcing (i.e., $\tau_c$), rather than CAPE, as being associated with variations of precipitation characteristics and forecast skill (Schwartz and Sobash, 2019).

Altogether, the ever changing meteorological situations in the Northwestern Mediterranean Coastal region are stratified into 48 strongly and 11 weakly forced days during HyMeX SOP1. All weakly forced, that is locally triggered precipitation events occur before mid-October with a sequence of weakly forced days in the beginning of September. This distribution follows the

seasonal cycle and reflects the climatological study of heavy precipitation events in northern Italy showing that weakly forced events occur from mid-May to the end of October with the highest frequency from mid-August to mid-September (Grazzini



et al., 2020). Key HyMeX IOPs are classified as strongly forced weather types in agreement with literature (Hally et al., 2014; Ducrocq et al., 2014; Nuissier et al., 2016). Likewise, the prominent heavy precipitation event that occurred during NAWDEX is clearly identified as strongly forced (Schäfler et al., 2018).

A clear connection between the weather regime and (i) the mean diurnal evolution of precipitation, (ii) the mean spatial distribution of daily rainfall, (iii) the precipitation predictability, (iv) the precipitation bias, (v) the probabilistic and (vi) spatial forecast quality is found. During strong synoptic control, which is dominating the weather on 80% of the days during HyMeX SOP1, the domain integrated precipitation predictability assessed with the normalized ensemble standard deviation $S_n$ is above average, the wet bias is smaller and the forecast quality is generally better. Conversely, there is a pronounced diurnal cycle of
area averaged precipitation and a considerable intra-ensemble variability in terms of placement of precipitation (i.e. large $S_n$) during weakly forced weather types consistent with previous results (e.g. Keil et al., 2019; Schwartz and Sobash, 2019; Bachmann et al., 2020). Disregarding the wet bias during weak control by focussing on 95th percentiles of precipitation shows the unexpected result of superior spatial predictability of most intense precipitation in the afternoon during weak control. We hypothesize that a reasonable representation of steady land surface structures (e.g. orography, coast line) in kilometric scale
numerical models provide trigger mechanisms to initiate convection during weak control and serve as a source of location predictability for precipitation, given favourable atmospheric conditions in this special geographical region. The important role of orography on precipitation in this region at this season is in agreement with the climatological study of Grazzini et al. (2020), who found that convective precipitation is largely influenced by orography during the frontal uplift with embedded equilibrium deep convection (herein: strong control) as well as non-equilibrium convection (herein: weak control). One reason
for the apparent overprediction of precipitation during weak control can partly be accounted for by the point type character of rain-gauge measurements that sample the spatial highly heterogeneous and intermittent nature of locally triggered convective precipitation insufficiently. This discrepancy calls for remotely sensed spatial rainfall measurements of high quality, that were not available in the present study.

It is shown that the unconditional evaluation of precipitation widely parallels the strongly forced weather type evaluation and
might obscure forecast model characteristics typical for weak control. Such a separation of statistics according to local weather conditions might proof useful to improve physical parameterisations that depend on the weather condition, as, for instance, Bouttier et al. (2012) suggested for the correlation lengths in the stochastic SPPT scheme, to identify a regime dependent impact of certain surface perturbations (Baur et al., 2018) or to enhance nowcasting capabilities (Kober et al., 2014).

*Acknowledgements.* The authors acknowledge access to the HyMeX database from which rainfall observations were retrieved. LC was
supported within the Transregional Collaborative Research Center SFB/TRR 165 "Waves to Weather" funded by the German Research Foundation (DFG). She was also supported by the Hans-Ertel-Centre for Weather Research. This German research network of universities, research institutes and Deutscher Wetterdienst is funded by the BMVI (Federal Ministry of Transport and Digital Infrastructure).



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
