# Peer review of "Dependence of Predictability of Precipitation in the Northwestern Mediterranean Coastal Region on the Strength of Synoptic Control"

_Atmospheric Chemistry and Physics, 2020_

## Referee Comment (RC1) · Anonymous Referee #1 · 9 Jul 2020

This is an interesting paper. The paper aims to use the convective adjustment timescale to examine the synopic control on convection in the Meditteranean region of France and Italy over a 2-month autumn period using the AROME convection-permitting ensemble, and hence determine the nature of the ensemble forecast predictability and performance in the different regimes.

The manuscript is largely clear and produces worthwhile results, but there are some aspects that need some further attention before it should be published. I'll first outline my main concerns before going into some other less crucial detail.

Main points:

[Figure]

1. You say that the domain should not have an area larger than 500x500km as recommended by Wernli and you do meet that criterion, as you say, but by having a domain 800km in the west-east direction are you not in danger of incorporating more than one larger-scale wave and therefore more than one regime (which you are trying to avoid)? Your third case (13th Oct) appears to have two areas of precipitation. One in the west that looks predictable and one further east that looks much less predictable. I wonder if you did your statistics for two overlapping domains each 500x300km whether you would get different results for that type of case and a better partitioning. Would it be possible to try that for at least that case? One of your main conclusions is that the strong forcing cases dominate, but is that partly because the domain is too extended and is always likely to capture a strongly forced event which will contribute the most to the timescale calculation?

2. Many times you say that the ensemble overpredicts the rain in the non-equilibrium cases, but I wonder how much of that is actually an artifact of the gauge interpolation missing rain than an over-prediction by the model. In a showery situation it is very likely the gauges will miss the heaviest rain cores, especially if locally focussed on hills where there are fewer gauges. A good test would be to take a model field, extract the values at the gauge locations and then do the interpolation. I would suspect you will get a lower domain-average value than the original field. Even if you don't try that out, it would still be worth mentioning in the article that a guage interpolation can miss rain when the rain coverage is low and has local spikes.

3. Some aspects of the methodolgy need a bit more clarity. You don't say why you choose a minimum rain amount of 3mm in 3h, which seems somewhat arbitary. A sentence or two about that would be helpful. I know you have a reference, but a few words would still be helpful. Linked to that, do you try to determine whether rain is convective or stratiform in nature? I assume you don't, but then there will sometimes be frontal rainbands that act to lower the timescale because rain occurs along with zero CAPE. You should at least mention this potential difficulty or explain why you

think it isn't a problem. I can see it making the convective timescale appear smaller especially going later into the year. Why do you choose 1mm/3h for the standard deviation, but 3mm/3h for the convective timescale calculation? What horizontal scale of Gaussian smoothing do you use? That could potentially have a singificant effect. If the smoothing is done over too large an area the precipitation threshold may not get exceeded anywhere in the scattered convection cases. On the other hand, some degree of smoothing will bring in more locations that may have high CAPE and that will affect (lengthen) the convective timescale. I take it, just for absolute clarity, you do not include any points with rain < 3mm/3h even if CAPE in non-zero?

4. To be honest I'm not sure about the value of some of the discussion of the individual cases It is useful to see the figures but there is a lot of descriptive text around them that isn't really adding much to the purpose of the paper, it's just describing where rain occurs in that event. A lot of readers will not know the location of regions in France, so it would be better to have some annotation on the figures to point to features instead. The key thing it seems is whether the rain is more or less widespread, and whether the members agree in say region "A" and region "B". Again, I'm not so convinced about the overestimation argument in weak control.

5. You should explain what you mean by the "ensemble mean FSS". Do you generate an ensemble mean precipitation field and then threshold for the FSS? That wouldn't be a good thing to do because you filter the true ensemble spread and change the frequency biases. Do you calculate the FSS for each member based on those thresholds and then take the average? Again that would not be the best thing to do because you might penalise ensemble spread as much as ensemble error. Do you threshold each member then take the ensemble mean of the binary probabilities and then apply the FSS? That would be the most sensible of those three options because it is evaluating the final probability field without clipping the distribution. Maybe you do something else? Definately it was good to choose the 95th percentile. Have you looked at other percentiles?

Other points:

1. You choose a threshold timescale of 3h, but then say it is different for summer and autumn (understandably). If you are most interested in partitioning out the days with the strongest and weakest synoptic control for evaluation could you just take the highest 30% and lowest 30% and then not have to worry about a timescale threshold. I'm not suggesting you do that here (as the results would be very similar), but it may help further studies of this sort.

2. When you talk about a "barrier" I assume you mean a stable layer? Sometimes storms form over mountains because of elevated heating and there isn't a clear inversion or organised storms form where there is an inversion but lifting mechanisms reduce it.

3. In some ways this seems to come down to wther the precipitation is contigouous or broken. I wonder if you were to classify that way whether you'd get something similar?

4. What would a graph of rain against Tau look like?

5. Figure 2 caption is hard to follow.

6. Might be interesting to know something about the spread of Tau and CAPE. Not suggesting you have to include that though.

7. You specify rank correlation values, but not actual correlation values. That leaves me a bit suspicious that they are not as good (closer to zero). Is that the case?

8. In line 163 you say "provides a better suitable measure" - but better than what?

9. For the indivuidual cases a few pressure contours would be informative to help set the context for the precipitation.

10. Presumably the skill-spread scores are picking up the bias, but also indicating potentially that there are too few members when evaluated at the grid scale, as well as saying the spread isn't sufficient.

11. I'm not sure it is a huge surprise that convection linked to mountains is more spatially predictable than convection that is mobile. I wonder what would happen if you just fixed a domain over the Alps and compared that with a flatter region for the convective timescale partitioning?

12. It might be worth also mentioning the paper by Flack et al Flack, D.L.A., Plant, R.S., Gray, S.L., Lean, H.W., Keil, C. and Craig, G.C. (2016), Characterisation of convective regimes over the British Isles. Q.J.R. Meteorol. Soc., 142: 1541-1553. doi:10.1002/qj.2758 This also references the papers you reference prior to 2016 along with Keil and Craig 2011, which you don't reference - which is a surprise.

There are some places where the text could be clearer (although in general it is well structured and readable), but it would probably be better to address these after dealing with the points above, which are going to involve changes in the text.

---

## Referee Comment (RC2) · Anonymous Referee #2 · 10 Jul 2020

General Comment

The paper discuss the performance of the AROME-EPS ensemble forecast for precipitation during the SOP of the Hymex Project, in dependency of the predictability of the events, as quantified by the convective adjustment timescale. The argument is scientifically very relevant, addressing the convective-scale predictability of the precipitation for an area interested by severe weather events. The work is well structured and meaningful, and clearly presented. However, I am not convinced of some conclusions, due to the verification process. I think there are some weaknesses in the verification interpretation, which hamper the conclusions to be drawn. Therefore, I recommend to

address some issues (described below), in particular in Section 5, before publishing the work.

Detailed Comments

Section 2.3 – Please add a reference for the Relative Operating Characteristics ROC and the reliability diagram. Though a description of these well know tool is not needed in the paper, not all the readers may be familiar with their definition. Figure 2: The thin lines are for me unreadable, and particularly their colour. Is it possible to increase the thickness?

Page 8: A small typo: "southeastern foothills if the Massif Central" should be "of"

Section 4 – In figure 6 also the spread of the ensemble is shown, by showing the area average precipitation of the members. There is evident that in the first case the spread is relatively "high", the members being quite different, as also noticed in the discussion of figure 7(b). In the second case, the spread of the precipitation is low, only 2 members having almost no rain, while all the others are close to each other. However, the first case is a predictably one, and the second a less predictable one. The spread, in the case of the precipitation, is not a good indicator, because it depends too much on the amount of precipitation itself: the first case is a predictable one, even if the members are different, because the rain is intense and the differences do not affect the "general performance" of the forecast. I think that, if the ensemble spread is shown, these considerations have to be made explicitly, otherwise the reader may receive a wrong message about the predictability. On top, the spread may be low even when the case is not well predicted, in case the ensemble is overconfident, which seems to be the case of the second case. For this reason, it would be good to have also the average observed precipitation, in figure 6.

Figure 7 and related discussion: an overprediction over Genova is noted in the 6-h period. Is this an overprediction in absolute sense or a timing problem (e.g. heavy precipitation occurred over Genova in the successive 6 hours?) Figure 11: the mean

of the RMS error of the ensemble members is shown and compared with the ensemble spread. Why is not shown instead the RMS error of the ensemble mean, which is the quantity which should be matched (statistically) by the spread? The chosen quantity is for sure higher than the other one, since the ensemble mean has (statistically) lower RMSE than all the members. Can you motivate a bit more the sentence (pag. 15): "The larger distance of the ROC curve points during strong control indicates the higher absolute spread when 3-hourly (and daily) precipitation accumulations are averaged over the entire SOP1."? Is this related to the point I raised about Section 4?

Page 15, about the sentence: "the forecast probabilities are consistently too large relative to the conditional observed relative frequencies. This is an indication of overforecasting equivalent to a wet bias.". The overforecasting in probability/frequency does not indicate a bias in the quantity, but in the probability. Therefore it does not indicate a wet bias, but an overconfidence of the ensemble. The members forecasting an event (e.g. 3mm/3h) are "too many" (therefore producing a too high probability of occurrence) with respect to the observed "probability" (which is the frequency) of occurrence of that event in the sample. I believe that the same overconfidence applies also to the dry areas (here you are considering only the wet areas, since you have a threshold >3mm/3h) and it is not related to an overestimation in the quantity itself. Page 17, from line 315 to the end of the Section. I am not convinced by the conclusions drawn here by the authors. "Taking this bias into account by using precipitation percentiles results in a superior spatial forecast quality during weakly forced regimes (Fig. 14b). Thus forecasting the location of heaviest precipitation in the afternoon (expressed by the 95th percentiles) is better during comparably quiescent synoptic-scale atmospheric conditions. This is at first sight an unexpected and surprising result." It is not the scattered nature of the weakly-forced precipitation field, when the isolated intense precipitation spots are selected, which gives an impression of skill by upscaling, just because somewhere a spot of precipitation is always available? I do not think that it is possible to conclude that there is a higher quality in the spatial forecast in case of weakly-forced cases based on this result with the FSS. I agree that orography "keeps"

the precipitation in place in case of convection, but I am not sure that with this increase of FSS for the 95th percentile can be a prove of skill.

---

## Author Comment (AC1) · 6 Oct 2020

Dear Editor, dear reviewers, please find attached the point-by-point response and the track-changed manuscript. We hope that the paper is now acceptable. Yours faithfully, Christian Keil

Please also note the supplement to this comment: https://acp.copernicus.org/preprints/acp-2020-508/acp-2020-508-AC1-supplement.zip

---

## Author Response (AR1)

MIM · Theresienstr.37 · 80333 München

Dr. Christian Keil

Telefon  +49 (0)89 2180-4447
Telefax  +49 (0)89 280 55 08

Christian.Keil@lmu.de

Meteorologisches Institut
Theresienstr. 37
80333 München
GERMANY

The Editor
Heini Wernli
ACP
Hymex Special Issue

München, 06/10/2020

**Revised Manuscript: acp-2020-508**
*Dependence of Predictability of Precipitation in the Northwestern Mediterranean Coastal Region on the Strength of Synoptic Control*

by C. Keil et al.

Dear Editor,

please find attached a revised version of the above submission to ACP. We are grateful for the positive and constructive criticism that the reviewers made on our submission. With their help, we were able to further improve our manuscript. A point-by-point response to the reviewers' comments is included below.

In the track-changed manuscript any revisions are indicated by changing the font colour to blue to show the modifications made in the course of the revision. We hope that the paper is now acceptable.

Yours faithfully,

Christian Keil

[Figure]

[Figure]

[Figure]

**Reply to Reviewers' comments**

**Reviewer 1**

*This is an interesting paper. The paper aims to use the convective adjustment timescale to examine the synoptic control on convection in the Mediteranean region of France and Italy over a 2-month autumn period using the AROME convection permitting ensemble, and hence determine the nature of the ensemble forecast predictability and performance in the different regimes.*

*The manuscript is largely clear and produces worthwhile results, but there are some aspects that need some further attention before it should be published. I'll first outline my main concerns before going into some other less crucial detail.*

*Main points:*

*1. You say that the domain should not have an area larger than 500x500km as recommended by Wernli and you do meet that criterion, as you say, but by having a domain800km in the west-east direction are you not in danger of incorporating more than one larger-scale wave and therefore more than one regime (which you are trying to avoid)?Your third case (13th Oct) appears to have two areas of precipitation. One in the west that looks predictable and one further east that looks much less predictable. I wonder if you did your statistics for two overlapping domains each 500x300km whether you would get different results for that type of case and a better partitioning.  Would it be possible to try that for at least that case?  One of your main conclusions is that the strong forcing cases dominate, but is that partly because the domain is too extended and is always likely to capture a strongly forced event which will contribute the most to the timescale calculation?*

The key point in the tau_c calculation is to choose a domain size for which the classification of the synoptic control is representative. As already written in the text '*The choice of the location and size of the investigation domain is carefully chosen and represents a compromise between being large enough to have numerous precipitation events giving good statistics, but small enough to comprise a specific and unambiguous meteorological situation in combination with the good coverage of rainfall observations in the Northwestern Mediterranean. If the domain is too large strongly differing meteorological systems may be contained and the results obtained using area averages may be blurred and not representative. However, we believe that the chosen domain … represents a good compromise being at the scale of the Rossby radius of deformation.*' Even before the original submission of the manuscript we divided the domain, as you suggest, into a French (West) and Italian (East) domain, but did not find important differences and decided to use this single Northwestern Mediterranean domain for the present study. Following your suggestion we repeated this for the overlapping domains covering the entire period, but did not find noteworthy differences. For instance, the tau_c values for the third case you mention amount to less than 30 min for all three

domains (NW Med, Western, Eastern). But you are right, if the domain is too large, i.e. larger than the Rossby radius of deformation (roughly 1000km), then you might mis-classify the predominant synoptic control.

Action: None.

*2. Many times you say that the ensemble overpredicts the rain in the non-equilibrium cases, but I wonder how much of that is actually an artifact of the gauge interpolation missing rain than an over-prediction by the model. In a showery situation it is very likely the gauges will miss the heaviest rain cores, especially if locally focussed on hills where there are fewer gauges. A good test would be to take a model field, extract the values at the gauge locations and then do the interpolation. I would suspect you will get a lower domain-average value than the original field. Even if you don't try that out, it would still be worth mentioning in the article that a guage interpolation can miss rain when the rain coverage is low and has local spikes.*

You are right, rain-gauges are prone to miss the heaviest rain cores in convective weather situations. Here a blend with radar observations yields a more realistic picture (taking into account all the uncertainties inherent in radar observations). We are totally aware of this shortcoming, and wrote in the final section of the original manuscript '*One reason for the apparent overprediction of precipitation during weak control can partly be accounted for by the point type character of rain-gauge measurements that sample the spatial highly heterogeneous and intermittent nature of locally triggered convective precipitation insufficiently. This discrepancy calls for remotely sensed spatial rainfall measurements of high quality, that were not available in the present study.*' To compute the conventional measures RMSE, ROC and reliability we used the nearest neighbour method and extracted the forecast values at the gauge location. In Fig. R1 we show the difference of taking the mean of the model field or the nearest neighbour forecast, respectively, and do not find important differences for the weakly forced case.

[Figure]

**Figure R1**: As Fig.6b in the manuscript, additionally depicting the ensemble mean precipitation based on the different calculations (Precip vs Precip NN Ensemble mean).

Action: We added in subsection 2.2. the sentence: We are aware that pointwise rain-gauge measurements can miss rain when the rain coverage is low and has local spikes, typically for weakly forced convective situations.

*3. Some aspects of the methodolgy need a bit more clarity. You don't say why you choose a minimum rain amount of 3mm in 3h, which seems somewhat arbitary. A sentence or two about that would be helpful. I know you have a reference, but a few words would still be helpful. Linked to that, do you try to determine whether rain is convective or stratiform in nature? I assume you don't, but then there will sometimes be frontal rainbands that act to lower the timescale because rain occurs along with zero CAPE. You should at least mention this potential difficulty or explain why you think it isn't a problem. I can see it making the convective timescale appear smaller especially going later into the year. Why do you choose 1mm/3h for the standard deviation, but 3mm/3h for the convective timescale calculation? What horizontal scale of Gaussian smoothing do you use? That could potentially have a singificant effect. If the smoothing is done over too large an area the precipitation threshold may not get exceeded anywhere in the scattered convection cases. On the other hand, some degree of smoothing will bring in more locations that may have high CAPE and that will affect (lengthen) the convective timescale. I take it, just for absolute clarity, you do not include any points with rain < 3mm/3h even if CAPE in non-zero?*

The rationale of taking 3mm/(3h) stems from earlier work (e.g. Keil et al. 2014, Kühnlein

et al. 2014) using a rain rate of 1 mm/h in raw, deterministic forecasts of convective scale models to separate rainy from non-rainy gridpoints in the tau_c calculation, since dry gridpoints preclude the computation. Due to the availability of 3-hourly data only, we temporally upscaled this threshold value. And yes, no gridpoints below this threshold value are used in the tau_c computation. On the other hand, there is a threshold needed to normalize the STDDEV of precipitation. This threshold is applied on the ensemble mean precipitation, not on individual members. Given the intra-member variability in precipitation forecasts we chose 1mm/(3h) to perform this normalisation.

Yes, the horizontal scale of Gaussian smoothing can have significant effects. Since tau_c represents an environment in which convection occurs, it is necessary to smooth the fields prior to the calculation. In the present work using convective scale models we kept the kernel with a half-width size of 20 GP (50 km) to stay consistent with the body of existing literature in which tau_c has been applied. In this study there is no additional distinction into stratiform or convective nature of precipitation, as was previously done based on CAPE values in e.g. Kober et al. 2014 and Grazzini et al. 2020, since precipitation is predominantly convective, and even frontal rainbands have considerable convective elements in the autumn season in the target region.

Action: The methodology is clarified in Section 2.

*4. To be honest I'm not sure about the value of some of the discussion of the individual cases It is useful to see the figures but there is a lot of descriptive text around them that isn't really adding much to the purpose of the paper, it's just describing where rain occurs in that event. A lot of readers will not know the location of regions in France, so it would be better to have some annotation on the figures to point to features instead. The key thing it seems is whether the rain is more or less widespread, and whether the members agree in say region "A" and region "B". Again, I'm not so convinced about the overestimation argument in weak control.*

We partly agree and reduced the descriptive text as well as some of the geographical terms in the text. However, we disagree that using region A vs region B increases readability and only kept key geographical terms shown in Figure 1.

Action: Text adapted throughout Section 4.

*5. You should explain what you mean by the "ensemble mean FSS". Do you generate an ensemble mean precipitation field and then threshold for the FSS? That wouldn't be a good thing to do because you filter the true ensemble spread and change the frequency biases. Do you calculate the FSS for each member based on those thresholds and then take the average? Again that would not be the best thing to do because you might penalise ensemble spread as much*

*as ensemble error.  Do you threshold each member then take the ensemble mean
of the binary probabilities and then apply the FSS? That would be the most
sensible of those three options because it is evaluating the final probability
field without clipping the distribution. Maybe you do something else?
Definately it was good to choose the 95th percentile.  Have you looked at
other percentiles?*

This is a good point and we are grateful that you raised this issue. Originally we calculated the FSS of each member before averaging. Now we follow your recommendation and apply the FSS on the ensemble mean of the binary probabilities. This results in slightly lower FSS values in Figure 13. Inspecting the 95th percentiles in Figure 14, there is an even clearer difference between both weather regimes. In particular, the size of the boxes during weak control is reduced.

Yes, we looked at other percentiles and present the results for the 85th percentiles in Figure R2. There is still a clear, albeit somewhat smaller distinction in spatial forecast quality between both. In particular the variability during strong control increases. Note that there are very low values for the 85th percentiles of observed precipitation during weak control. The use of even lower percentiles is limited by the spatial extent of precipitation during weak control.

[Figure]

**Figure R2**: As Fig.14 in the manuscript but for 85th percentiles of precipitation.

Action: New Figures 13 and 14 and text adapted in Section 5.

*Other points:*

*1. You choose a threshold timescale of 3h, but then say it is different for
summer and autumn (understandably). If you are most interested in partitioning*

*out the days with the strongest and weakest synoptic control for evaluation
could you just take the highest 30% and lowest 30% and then not have to worry
about a timescale threshold. I'm not suggesting you do that here (as the
results would be very similar), but it may help further studies of this sort.*

Thank you for this interesting suggestion. We will apply this in future studies.

Action: None.

*2. When you talk about a "barrier" I assume you mean a stable layer? Sometimes
storms form over mountains because of elevated heating and there isn't a clear
in-version or organised storms form where there is an inversion but lifting
mechanisms reduce it.*

Yes, for instance a stable layer or the presence of convective inhibition.

Action: Clarified in the text.

*3. In some ways this seems to come down to wether the precipitation is
contingous or broken. I wonder if you were to classify that way whether you'd
get something similar?*

We guess so, and we are curious if you are aware of a suitable measure to classify
texture. In the past we used the fractional coverage as predictor, but this quantity does
not really describe the texture of the precipitation field.

Action: None.

*4. What would a graph of rain against Tau look like?*

This is shown in Figure R3. There is no new insight in view of a classification of weather
regimes apart the fact that there tends to fall more rain during strong control (i.e. small
Tau values).

[Figure]

**Figure R3**: Scatterplot of Tau and rain amount.

Action: None.

*5. Figure 2 caption is hard to follow.*

Ok.

Action: Clarified.

*6. Might be interesting to know something about the spread of Tau and CAPE. Not suggesting you have to include that though.*

We leave this for future work.

Action: None.

*7. You specify rank correlation values, but not actual correlation values. That leaves me a bit suspicious that they are not as good (closer to zero). Is that the case?*

Yes, these amount to 0.34. Since we are primarily interested in the pairing of data we chose the rank correlation, a statistical quantity describing exactly this (Wilks 2011) and waive to show the Pearson correlation value in the text.

Action: None.

*8. In line 163 you say "provides a better suitable measure" - but better than what?*

Better than, for instance, CAPE values.

Action: Clarified in the text.

*9. For the indivuidual cases a few pressure contours would be informative to help set the context for the precipitation.*

We presently don't have the data retrieved and think such isobars don't give new insights.

Action: None.

*10. Presumably the skill-spread scores are picking up the bias, but also indicating potentially that there are too few members when evaluated at the grid scale, as well as saying the spread isn't sufficient.*

Yes, we agree, a 12-member ensemble is at the lower edge in terms of ensemble size.

Action: None.

*11. I'm not sure it is a huge surprise that convection linked to mountains is more spatially predictable than convection that is mobile. I wonder what would happen if you just fixed a domain over the Alps and compared that with a flatter region for the convective timescale partitioning?*

Yes, we generally agree and changed the text (see reviewer 2 comments, too). In this region it is difficult to fix a flat domain that is large enough to be informative (see above arguments on the domain size).

Action: Clarified.

*12. It might be worth also mentioning the paper by Flack et al Flack, D.L.A., Plant,R.S., Gray, S.L., Lean, H.W., Keil, C. and Craig, G.C. (2016), Characterisation of convective regimes over the British Isles. Q.J.R. Meteorol. Soc., 142: 1541-1553.doi:10.1002/qj.2758 This also references the papers you reference prior to 2016 along with Keil and Craig 2011, which you don't reference - which is a surprise. There are some places where the text could be clearer (although in general it is well structured and readable), but it would probably be better to address these after dealing with the points above, which are going to involve changes in the text.*

Action: We added these references, thank you.

[Figure]

[Figure]

[Figure]

**Reviewer 2**

*The paper discuss the performance of the AROME-EPS ensemble forecast for precipitation during the SOP of the Hymex Project, in dependency of the predictability of the events, as quantified by the convective adjustment timescale. The argument is scientifically very relevant, addressing the convective-scale predictability of the precipitation for an area interested by severe weather events. The work is well structured and meaningful, and clearly presented. However, I am not convinced of some conclusions, due to the verification process. I think there are some weaknesses in the verification interpretation, which hamper the conclusions to be drawn. Therefore, I recommend to address some issues (described below), in particular in Section 5, before publishing the work.*

*Detailed Comments*

*Section 2.3 – Please add a reference for the Relative Operating Characteristics ROC and the reliability diagram. Though a description of these well know tool is not needed in the paper, not all the readers may be familiar with their definition.*

We added these references:

Wilks, D.S. (2011) Statistical Methods in the Atmospheric Sciences, 3rd edition. Elsevier Academic Press, Amsterdam, Netherlands.

Jolliffe IT, Stephenson DB. 2011. Forecast Verification: A Practitioner's Guide in Atmospheric Science (2nd edn). John Wiley and Sons: Chichester, UK, doi:10.1002/9781119960003.ch1

Action: Done.

*Figure 2: The thin lines are for me unreadable, and particularly their colour. Is it possible to increase the thickness?*

The thickness of the thin lines is increased.

Action: Done.

*Page 8: A small typo: "southeastern foothills if the Massif Central" should be "of"*

Thank you.

Action: Done.

*Section 4 – In figure 6 also the spread of the ensemble is shown, by showing the area average precipitation of the members. There is evident that in the*

*first case the spread is relatively "high", the members being quite different,*
*as also noticed in the discussion of figure 7(b). In the second case, the*
*spread of the precipitation is low, only 2 members having almost no rain,*
*while all the others are close to each other. However, the first case is a*
*predictably one, and the second a less predictable one. The spread, in the*
*case of the precipitation, is not a good indicator, because it depends too*
*much on the amount of precipitation itself: the first case is a predictable*
*one, even if the members are different, because the rain is intense and the*
*differences do not affect the "general performance" of the forecast. I think*
*that, if the ensemble spread is shown, these considerations have to be made*
*explicitly, otherwise the reader may receive a wrong message about the*
*predictability. On top, the spread may be low even when the case is not well*
*predicted, in case the ensemble is overconfident, which seems to be the case*
*of the second case. For this reason, it would be good to have also the average*
*observed precipitation, in figure 6.*

Thank you, we agree and completely revised Fig.6 that includes the average observed precipitation and the normalized ensemble spread $S\_n$ now as well. It is important to distinguish relative (normalized; $S\_n$) and absolute ensemble spread. In absolute terms there is a larger variability during strong forcing cases (due to higher rainfall intensities, see e.g. individual members in Fig.6c). The normalized spread $S\_n$ gives indication on predictability and is evidently larger for the second case pointing towards the below average predictability. Additionally the y-coordinate has been 'rescaled' to highlight the differences between the three prominent cases.

Action: New Figure 6 and corresponding discussion in Section 4.

*Figure 7 and related discussion: an overprediction over Genova is noted in the*
*6-hperiod. Is this an overprediction in absolute sense or a timing problem*
*(e.g. heavy precipitation occurred over Genova in the successive 6 hours?)*

Figure R4 indicates that there is no major timing error for IOP16a in this region. In the successive 6 hours (18-24 UTC) the rainfall amounts compare better in the proximity of Genova, whereas rainfall is overestimated further to the east.

[Figure]

**Figure R4**: Illustration of 6-hourly precipitation for IOP16a valid 26 October (left) 12-18 and (right) 18-24 UTC.

*Figure 11: the mean of the RMS error of the ensemble members is shown and compared with the ensemble spread. Why is not shown instead the RMS error of the ensemble mean, which is the quantity which should be matched (statistically) by the spread? The chosen quantity is for sure higher than the other one, since the ensemble mean has (statistically) lower RMSE than all the members.*

Thank you for pointing this out. Actually we computed the RMSE of the ensemble mean and corrected the text.

Action: Caption and text corrected.

*Can you motivate a bit more the sentence (pag. 15):"The larger distance of the ROC curve points during strong control indicates the higher absolute spread when 3-hourly (and daily) precipitation accumulations are averaged over the entire SOP1."? Is this related to the point I raised about Section 4?*

We clarified this point: "The larger distance of the ROC curve points from the diagonal (resulting in larger concavity) during strong control indicates greater event discrimination when 3-hourly (and daily) precipitation accumulations are averaged over the entire SOP1."

Action: Text changed.

*Page 15, about the sentence: "the forecast probabilities are consistently too large relative to the conditional observed relative frequencies. This is an indication of overforecasting equivalent to a wet bias.". The overforecasting in probability/frequency does not indicate a bias in the quantity, but in the probability. Therefore it does not indicate a wet bias, but an overconfidence of the ensemble. The members forecasting an event (e.g. 3mm/3h) are "too many" (therefore producing a too high probability of occurrence) with respect to the observed "probability" (which is the frequency) of occurrence of that event in the sample. I believe that the same overconfidence applies also to the dry areas (here you are considering only the wet areas, since you have a threshold >3mm/3h) and it is not related to an overestimation in the quantity itself.*

Thank you for this clarification, we appreciate your comment and agree.

Action: Text changed accordingly.

*Page 17, from line 315 to the end of the Section. I am not convinced by the conclusions drawn here by the authors. "Taking this bias into account by using precipitation percentiles results in a superior spatial forecast quality during weakly forced regimes(Fig. 14b). Thus forecasting the location of heaviest precipitation in the afternoon (ex-pressed by the 95th percentiles)*

*is better during comparably quiescent synoptic-scale atmospheric conditions. This is at first sight an unexpected and surprising result."* It is not the scattered nature of the weakly-forced precipitation field, when the isolated in-tense precipitation spots are selected, which gives an impression of skill by upscaling, just because somewhere a spot of precipitation is always available? I do not think that it is possible to conclude that there is a higher quality in the spatial forecast in case of weakly-forced cases based on this result with the FSS. I agree that orography "keeps" the precipitation in place in case of convection, but I am not sure that with this increase of FSS for the 95th percentile can be a prove of skill.

Thank you, however we disagree with your point. By taking the 95th percentiles of precipitation the 5% gridpoints are selected that receive the strongest precipitation, independent to the synoptic control. These upper 5% gridpoints are fairly scattered in both regimes. The FSS of the 95th percentiles gives only information on the position of the strongest precipitation, but *not* on intensities. Considering the 6h accumulated precipitation between 12-18 UTC (the 'convective period') the finding on superior *pure* spatial accuracy of the 95th percentiles coincides with common meteorological sense that deep convection and thunderstorms are strongly linked to orography during weakly forced situations. In contrast, the *timing and position* of heaviest rainfall during strong synoptic control is crucially linked to the cyclone track and only modulated to a minor degree by orography.

Action: We emphasize that we talk on the *pure* forecast location accuracy at various places in the entire manuscript when examining the FSS of the 95th percentiles of precipitation.

[revised manuscript text omitted]

---

## Author Response (AR2)

MIM · Theresienstr.37 · 80333 München

Dr. Christian Keil

Telefon +49 (0)89 2180-4447
Telefax +49 (0)89 280 55 08

Christian.Keil@lmu.de

Meteorologisches Institut
Theresienstr. 37
80333 München
GERMANY

The Editor
Heini Wernli
ACP
Hymex Special Issue

München, 26/10/2020

**Revised Manuscript: acp-2020-508 (2)**
*Dependence of Predictability of Precipitation in the Northwestern Mediterranean Coastal Region on the Strength of Synoptic Control*

by C. Keil et al.

Dear Editor,

please find attached a second revised version of the above submission to ACP. We are grateful for the positive and constructive criticism that the reviewers made on our submission. We carefully looked at grammar issues throughout the text and incorporated all points raised by Referee 1. With your help, we were able to further improve our manuscript.

We hope that the paper is now acceptable for publication.

Yours faithfully,

Christian Keil

[Figure]

[Figure]

[Figure]

MIM · Theresienstr.37 · 80333 München

The Editor
Heini Wernli
ACP
Hymex Special Issue

Dr. Christian Keil

Telefon  +49 (0)89 2180-4447
Telefax  +49 (0)89 280 55 08

Christian.Keil@lmu.de

Meteorologisches Institut
Theresienstr. 37
80333 München
GERMANY

München, 27/10/2020

**Revised Manuscript: acp-2020-508 (2)**
*Dependence of Predictability of Precipitation in the Northwestern Mediterranean Coastal Region on the Strength of Synoptic Control*

by C. Keil et al.

Dear Editor,

Here a point-by-point response you were asking for:

* Editor's comment on area size

Action: Added river catchment alternative in Section 2.2

* Referee 2 comments:

1) Manuscript carefully re-read focussing on grammar issues. E.g. added 'the' or 'a' at various places, changed 95th percentile without 's' throughout the text. Done.

2) Wording in abstract corrected. Done.

3) Added sentence like 'examined sub-domains and got similar results' in Section 2.2 and mention to stay below the Rossby Radius of Deformation O(1000km). Done.

3 and 4) Text accordingly corrected.

5) Added '(see Fig. 1 for geographical landmarks)' in Sections 3 and 4 to help the reader navigate in that area.

6) Added 'overconfidence could still come from a wet bias or a tendency to have too much spatial agreement.' in the text.

7) Inserted 'than' twice as reviewer suggested.

That's all.

Kind regards,

Christian